# Dual-Band Monopole MIMO Antenna Array for UAV Communication Systems

**DOI:** 10.3390/s24185913

**Published:** 2024-09-12

**Authors:** Muhammad Usman Raza, Hongwei Ren, Sen Yan

**Affiliations:** School of Information and Communications Engineering, Xi’an Jiaotong University, Xi’an 710049, China; usman_786@stu.xjtu.edu.cn (M.U.R.); rhw718@stu.xjtu.edu.cn (H.R.)

**Keywords:** dual-band, MIMO monopole antenna, UAV, communication system, sequential, impedance bandwidths, transmission efficiency

## Abstract

This study proposes a compact, low-profile, four-port dual-band monopole multiple-input-multiple-output (MIMO) antenna array for unmanned aerial vehicle (UAV) communication systems. Each monopole antenna of the array features a modified T-shaped radiator configuration and is printed on a Rogers RT5880 substrate with compact dimensions of 134.96 mm × 134.96 mm × 0.8 mm. A four-element square MIMO configuration with sequential 0°, 90°, 180°, and 270° rotations was integrated smoothly into the UAV body. A prototype of the MIMO array was fabricated and experimentally evaluated, with measured results showing a close correlation to simulated results. The proposed dual-band monopole antenna demonstrated one of the widest impedance bandwidths of 46.15% at 2.4 GHz (2.04 to 3.25 GHz) IEEE 802.11b and 31.85% at 5.8 GHz (5.37 to 7.38 GHz) IEEE 802.11a on a thin 0.0064 λ_o_ substrate while achieving high transmission efficiency. The isolation of the proposed four-port MIMO design was measured at 23 dB at 2.4 GHz and 19 dB at 5.8 GHz. The MIMO array’s total efficiency of each monopole antenna was measured at 96% at 2.4 GHz and 89% at 5.8 GHz. The design has measured diversity parameters such as an ECC below 0.01 and a DG of approximately 10. Based on these results, the proposed design suits the UAV communication system.

## 1. Introduction

The unmanned aerial vehicle (UAV) communication network system is a wireless infrastructure engineered specifically to connect UAVs, which are commonly called drones. These networks allow drones to reliably send and receive information among groups of UAVs in mid-air as well as enable data transmission for drone-to-drone in-flight and drone-to-ground transmissions [1,2]. The UAV communication system aims to establish stable, robust, and reliable networks for a variety of applications, including agriculture monitoring, fire fighting, chemical spraying, surveillance operations, disaster response, and delivery services, while also leveraging wireless power transfer (WPT) technology to improve the operational efficiency and prolong flight times without the physical connections [3,4,5,6,7,8,9,10,11,12]. By facilitating reliable communication and synchronization between groups of drones and ground control centers, these networks unlock advanced use cases for drone technologies. Drones networked using UAV communication systems can quickly assess damage following disasters for public safety. Delivery drones can operate the network to link with logistics centers for commercial purposes, allowing for the efficient transportation of items to consumer locations [13,14,15]. It is crucial to utilize antennas with larger bandwidths and higher efficiency to enhance data transmission over longer distances for UAVs. These antennas should be lightweight, thin, and conformal to seamlessly integrate with the UAV’s body without protrusion. The goal is to ensure a reliable and high data transmission system for UAV operations. One promising approach to achieve this goal is integrating MIMO technology with dual-band antenna arrays on UAVs. MIMO systems leverage multiple antennas at both the transmitter and receiver ends to transmit and receive multiple data streams simultaneously, thereby increasing data rates, spectral efficiency, and overall system performance [16]. By deploying MIMO antenna arrays on UAVs, these aerial platforms can serve as efficient 5G communication hubs, facilitating reliable and high-speed data transmission between ground users and the UAV-based base station, as depicted in Figure 1. The dual-band capability of the MIMO antenna array allows for operation in multiple frequency bands, such as the widely used 2.4 GHz and 5 GHz bands for Wi-Fi [17,18] and other wireless applications, providing additional bandwidth and frequency diversity. In this scenario, ground users can use MIMO technology to communicate with the UAV-based MIMO antenna array, benefiting from increased data rates, improved spectral efficiency, and enhanced link reliability [19]. The UAV-based MIMO antenna array can also act as a relay, receiving data from ground transmitters and forwarding it to ground users or other network nodes. This relay functionality is particularly useful in scenarios where direct communication between ground users is hindered by obstacles, terrain, or distance limitations [20,21]. Furthermore, integrating MIMO technology with dual-band antenna arrays on UAVs addresses the challenges of size and weight constraints, as well as the dynamic nature of UAV operations. Compact and lightweight antenna designs, advanced signal processing algorithms for channel estimation and beamforming, and innovative antenna array configurations can be employed to optimize the performance of the MIMO system in the dynamic UAV environment. Numerous single-band, dual-band, single-antenna, and MIMO antennas have been proposed in recent decades. While the proposed single broadband antenna operates in a 3 GH to 4.30 GHz frequency band, as reported in [22], it does not cover the Wi-Fi bands. This limitation highlights the potential for integrating MIMO technology in future antenna designs to improve frequency coverage and overall performance. Blade antennas [23,24], annular ring slot antennas [25], microstrip patch antennas [26], slotted blade dipole antennas [27], conformal beamwidth antennas [28], conformal monopolar UAV antennas [29], and flush-mounted monopolar patch UAV antennas with large ground planes [30] are among the commonly utilized antenna types for UAVs. However, these single antennas offer lower bandwidth and limited radiation coverage. A quad-port circular spanner-shaped MIMO antenna array with defected ground structure at ground configuration [31] and a two-port microstrip MIMO antenna [32,33] have been previously reported for UWB and WLAN applications, respectively. However, these designs did not address the 2.4 GHz frequency band. UAVs frequently utilize communication antenna arrays to link with ground stations, as shown in Figure 2. Figure 2 shows an illustration of the radiation coverage analysis for a UAV equipped with a MIMO antenna array. The MIMO antenna array, illustrated for two frequency bands, offers radiation coverage in numerous UAV positions relative to the ground station. The MIMO antenna array offers broader and more constant radiation patterns across different scenarios, whether the drones are directly overhead (Drone 1), trajectory location (Drone 2), or far away (Drone 3) from base stations. The dual-band radiation pattern coverage allows the MIMO antenna arrays to maintain robust communication links even as the drone’s position changes. This contrasts with single antennas with relatively narrower radiation patterns, susceptible to signal degradation and struggling to maintain a connection at different distances and orientations.

In this study, a proposed microstrip dual-band monopole MIMO antenna array is measured on a thin substrate with a thickness of 0.0064λo, which excites two bands, i.e., 2.4 GHz IEEE 802.11b and 5.8 GHz IEEE 802.11a. The dual-band monopole MIMO antenna demonstrates outstanding characteristics such as a broad bandwidth and high efficiency, which are thoroughly analyzed herein. The dual-band operation led to the generation of two distinct radiation patterns, as illustrated in Figure 2. This characteristic is particularly advantageous in scenarios where a single radiation pattern, typical for a single band, could lead to coverage issues. The research presented in this paper presents an analysis of the evolution of the antenna design and conducts a sensitivity analysis of key design parameters affecting the surface current distribution. A 2 × 2 MIMO array prototype was fabricated and measured to validate the design. Simulations were performed using CST Microwave Studio 2021, and measurements were conducted in an anechoic chamber. This design enables high data rates for reliable UAV 5G communication links.

## 2. Monopole Antenna Design and Configuration

The proposed dual-band monopole antenna’s geometry was specifically designed to achieve the broadest bandwidth possible and tailored to meet the specific requirements of UAV communication systems where the antenna array functions as a radiating element. As depicted in Figure 3, the antenna’s geometry was carefully designed based on a parametric study optimized through computer s-parameter simulation in CST studio. The monopole antenna was designed using Rogers RT5880 with a dielectric constant of 2.2 and a thickness of 0.8 mm, approximately 0.0064λo at 2.4 GHz. Rogers RT5880 (Building A, Shixia Ganglian Industrial Park, Shenzhen, China) is a ceramic-filled PTFE composite material with a precise tolerance and a low dielectric constant, making it suitable for various high-frequency applications. It is sufficiently cheap and readily available on the market. This material has been used in antenna designs operating at frequencies of up to 45 GHz. The designed antenna’s proposed dual-band radiation monopole patch was integrated with a rectangular microstrip feed line. This microstrip feed line, characterized by an L7 length and L9 width, was strategically printed between the two ground planes. The ground planes themselves are defined by a parametric length of L5 and width of L6. The feed line was fed with a 50 Ω SMA connector for signal transmission. This configuration ensures efficient energy transfer between the radiating patch and the associated circuitry. The precise parameters of each antenna component play a crucial role in defining its performance characteristics; the antenna’s geometric dimensions are detailed in Table 1.

The proposed dual-band monopole antenna was designed to achieve dual-band functionality for UAV applications through a modified radiation patch. The proposed design evaluation process is depicted in Figure 4. Stage 1, as illustrated in Figure 4a, is a planar rectangular monopole antenna dimensions of 18 mm × 28.45 mm. The length of the ground plane affects the dual-band resonating bandwidth. The s-parameter S11 result is above −10 dB and therefore within the desired dual-frequency band of 2.4 GHz and 5.8 GHz, which indicates that the monopole antenna is not well-matched, as shown in Figure 5, Stage 1. In Stage 2, the rectangular monopole antenna was etched on the side by adding slot I on the left side, as shown in Figure 4b. This adjustment resulted in an S11 value below −10 dB, and therefore within the desired dual-frequency band from 2.21 GHz to 3.36 GHz and 5.2 GHz to 7.6 GHz, as depicted in Figure 5, indicating that the monopole dual-band antenna is well-matched. However, it did not achieve the desired bands. Stage 3, as shown in Figure 4c, is the proposed T-shaped monopole antenna. The S11 was below −10 dB, and therefore within the desired dual-band from 2.24 GHz to 3.53 GHz and 5.53 GHz to 7.98 GHz, as depicted in Figure 5, indicating that the dual-band monopole antenna is well-matched. Additionally, both bands were slightly moved to higher frequencies, improving the impedance bandwidth of the desired dual bands of 2.4 GHz and 5.8 GHz. These simulations were conducted using CST Studio to evaluate the antenna’s performance.

The study identified and selected the dimension that exhibited the most favorable behavior by simulating antennas for different optimized dimension values and analyzing the results. The S-parameter S11 results obtained by modifying dimensions L4 and L5 of the monopole antenna are depicted in Figure 6. The analysis revealed a significant influence of these dimensions on the antenna’s response, with the optimal results achieved at L4 = 11.8 mm and L5 = 27.5 mm. Notably, variations in L4 and L5 led to frequency band shifts toward higher frequencies, as illustrated in Figure 6a,b. Simulated −10 dB impedance bandwidths of 46.15% at 2.4 GHz IEEE 802.11b and 31.85% at 5.8 GHz IEEE 802.11a were noticed, respectively. The study emphasizes the critical role of dimension optimization in achieving desired antenna performance characteristics for UAV communication systems. By carefully adjusting parameters such as L4 and L5, the antenna’s frequency response can be tailored to meet specific bandwidth requirements, ensuring efficient communication within the designated frequency bands.

The surface current distribution on the proposed dual-band monopole antenna at the frequencies of 2.40 GHz IEEE 802.11b and 5.8 GHz IEEE 802.11a [34], as depicted in Figure 7, shows that the current was concentrated around the microstrip feed line and distributed along the antenna edges, with a maximum surface current of 14.8 A/m. Moving toward the center of the patch, the current density gradually decreased. Monopole antennas consist of a straight rod-shaped conductor mounted perpendicularly over a conductive surface, with the driving signal applied between the lower end of the monopole and the ground plane.

The equivalent circuit model of the proposed dual-band monopole antenna is depicted in Figure 8. This simplified representation comprises three main components: a feed network with a coaxial port and two resonant circuits, as shown in Figure 8b. The feed network, which includes the SMA connector Zp, Cf, Lf, and Rf, is essential for signal transmission, impedance matching, and minimizing reflection and power loss. Two parallel RLC circuits were used to simulate the dual-band monopole antenna’s behavior [35,36]. The lower resonance at 2.4 GHz is represented by the first RLC circuit (La1, Ca1, Ra1), while the higher resonance at 5.8 GHz is defined by the second RLC circuit (La2, Ca2, Ra2). Each RLC circuit contains the resonant characteristics of the dual-band monopole antenna at its respective frequency band, including the bandwidth and quality factor. This equivalent circuit approach enables a comprehensive analysis of the dual-band monopole antenna’s electrical behavior across its resonating frequency bands of 2.4 GHz and 5.8 GHz. The antenna’s performance, bandwidth characteristics, and matching impedance can be fine-tuned by optimizing the values of these lumped elements. The accuracy of the proposed dual-band monopole antenna structure was confirmed by the strong correlation between the simulated results and the simulated performance of the equivalent circuit, as shown in Figure 9. This correlation demonstrates the model’s capability to effectively represent the monopole antenna’s intricate electromagnetic behavior using basic lumped elements. The proposed dual-band monopole antenna’s structure and its equivalent circuit were validated via simulation.

## 3. Proposed MiMO Antenna Array Configuration

The proposed configuration employs four dual-band monopole antennas and positions them in a square pattern, as illustrated in Figure 10a. These monopole antennas were placed at the square’s four corners with center-to-center distance d = 0.5λ_o_ between each one. These are antennas A through D, with A being rotated at 0°, B at 90°, C at 180°, and D at 270° to enhance the performance. Another advantage of the standard UAV design of 2 × 2 MIMO dual-band monopole antennas at the UAV’s center is that it ensures a symmetrical radiation pattern, as shown in Figure 10b. By residing in the middle of the framework, this method effectively enables consistent communication in all directions and facilitates omnidirectional coverage, which are essential for the MIMO systems. Due to the positioning of the antennas at the center of the UAV frame, the distance from potential sources of electromagnetic interference (EMI) is maximized. It appears that this higher level of electrical isolation is beneficial in reducing any EMI produced by motors, blades, or any of the onboard electronics, which could otherwise degrade antenna performance. The central position for the installation also entails antennas with an unobstructed line of sight in all directions. This line of sight is necessary for maintaining consistent link quality during flight, especially in MIMO techniques that utilize spatial diversity. This configuration ensures more reliable communication links by reducing the signal blockage from the UAV frame or other components. UAVs often operate in dynamic environments, where the orientation of the communication link can very quickly. The design with a centrally positioned dual-band monopole MIMO antenna array offers more uniform radiation coverage and, therefore, is able to satisfy such variation in the link requirements. This placement approach improves the system’s ability to handle flight scenarios and preserve robust connections. The dual-band monopole antenna significantly improved the MIMO system’s capacity to manage multi-path propagation. This enhancement led to increased data rate and overall link robustness, which are critical factors in UAV communications.

Figure 11 shows the s-parameter measurement setup for the fabricated prototype of the 2 × 2 MIMO dual-band monopole antenna array designed for UAV communications. Figure 11a displays a 2 × 2 MIMO monopole antenna array, where the four monopole antennas are printed as “1”, “2”, “3”, and “4” on the substrate. One of the antenna ports was connected to an RF coaxial cable, which was used to feed the signal into the antenna. The remaining three ports were connected to loads. These loads likely act as terminators to block signal reflections that could interfere with the measurements, thus producing valid and reliable results of the antenna’s performance.

Figure 11b demonstrates the measurement setup using an Agilent AV3672E-S vector network analyzer (VNA), an essential instrument for assessing the s-parameters of the antenna array. VNA uses an LCD screen to display the measurement results in real time, typically in the form of s-parameter plots. The RF cables connect the 2 × 2 MIMO dual-band monopole antenna array to the VNA, facilitating the transmission and reception of signals for analysis. The VNA is then calibrated to ensure accurate measurements, which entails connecting standards such as open, short, and load and compensating for any systematic errors. While measuring, the VNA applies a known signal through the connected port and acquires the reflected and transmitted signals. This is carried out for each port to ensure that enough data have been obtained concerning the antenna’s performance. These collected s-parameters are then used to define such parameters as return loss and mutual coupling between the antenna elements.

The simulated and measured results, provided in Figure 12, were evaluated at 2.4 GHz IEEE 802.11 and 5.8 GHz IEEE 802.11, as shown in Figure 12a. Across both bands, the reflection coefficients indicated consistent −10 dB bandwidths and sufficient isolation values higher than 20 dB between elements. The reflection coefficients for monopole Ant-A and Ant-B exhibited identical values at a center-to-center spacing of 0.5λ_o_. The transmission coefficients between neighboring elements, specifically S21 and S23, were below −28 dB across the 2.4 GHz band (2.04 to 3.25 GHz) and below −34 dB across the 5.8 GHz band (5.37 to 7.38 GHz), as shown in Figure 12b. Similarly, the transmission coefficients between diagonal elements S31 and S42 were below −23 dB at 2.4 GHz and −19 dB at 5.8 GHz, as shown in Figure 12c. Additionally, the transmission coefficient S41 was below −28 dB at 2.4 GHz and −34 dB at 5.8 GHz, as shown in Figure 12d. The results from simulations and measurements usually align closely. However, minor discrepancies between the measured and simulation results may arise from issues such as incorrect fabrication or poor soldering of SMA connectors.

Figure 13 shows a detailed measurement setup to test the radiation pattern of the 2 × 2 MIMO dual-band monopole antenna designed inside a shielded anechoic chamber. The anechoic chamber provides the necessary protection from outside electromagnetic interference, thus offering accurate measurement. The system also has a transmitting (Tx) base antenna that transmits test signals to the UAV dual-band MIMO antenna array that is mounted on a rotational platform to enable orientation in different directions. These measurements were carried out in the far-field region from the Tx antenna and the antenna under study so that one can assume plane wavefronts. The antenna array of the UAV can tilt on both horizontal (θ-axis) and vertical (φ-axis) planes, as this enables measurements along the radiation pattern. It also provides provisions for the signal polarizations by specifying the θ-polarization and φ-polarization parts of it. To connect the Tx antenna and the antenna under study, RF cables were used and connected to an Agilent AV3672E-S Vector Network Analyzer (VNA) (Aviatronik, Roma, Italy) to measure the performance parameters, such as the s-parameters of the antenna across the various frequencies. A computer managed the measurement process and also collected data from the VNA and performed calculations of the provided evaluation, including the radiation pattern and other metrics.

The antenna prototype’s radiation patterns were assessed in a microwave anechoic chamber, as depicted in Figure 14. The E-plane and H-plane are key testing planes for evaluating the antenna’s radiation pattern. Within the chamber, these planes enable study of the radiation pattern, providing insights into the antenna’s electromagnetic behavior. Testing in this controlled environment of the anechoic chamber is vital for optimizing the dual-band MIMO monopole antenna array’s performance and ensuring precise assessment, free from external interference. The simulated and measured normalized far-field radiation patterns in the H-plane and E-plane at 2.4 GHz and 5.8 GHz are presented in Figure 15. The proposed antenna exhibited a dipolar pattern in the E-plane and a near-omnidirectional pattern in the corresponding H-plane. Sequential rotation of a 2 × 2 dual-band monopole MIMO antenna array was implemented to provide efficient data transmission to the ground station and comprehensive coverage. The dipolar pattern in the E-plane shows the antenna’s directional properties, whilst the near-omnidirectional pattern in the H-plane suggests that it may radiate energy more evenly over a larger angle. The 2 × 2 MIMO dual-band monopole antenna array’s sequential rotation improved data transmission efficiency and communication dependability by providing good coverage in all directions.

The simulated and measured total efficiency discrepancies for Ant-A are highlighted by the results shown in Figure 16, which are a scribed to high isolation levels and excellent impedance matching. Ant-A’s low profile and excellent isolation capabilities allow it to attain overall efficiencies of over 96% at 2.4 GHz IEEE 802.11 and over 89% at the 5.8 GHz IEEE 802.11 bands. The outstanding performance of the dual-band monopole Ant-A in maintaining robust signal transmission capabilities across the designated frequency bands is highlighted by these high-efficiency figures, providing dependable communication in the UAV system.

The various diversity parameters examined include the envelope correlation coefficient (ρ) and diversity gain (*DG*), which together characterize the typical performance of MIMO antennas [37,38,39]. This be calculated as
(1)$$\rho =\frac{{\left|\iint_{4\pi }\left({\vec{R}}_{i}\left(\theta ,\varphi \right)\right)\times \left({\vec{R}}_{j}\left(\theta ,\varphi \right)\right)d\Omega \right|}^{2}}{\iint_{4\pi }{\left|\left({\vec{R}}_{i}\left(\theta ,\varphi \right)\right)\right|}^{2}d\Omega \times \iint_{4\pi }{\left|\left({\vec{R}}_{j}\left(\theta ,\varphi \right)\right)\right|}^{2}d\Omega }$$
where Ω indicates the solid angle in 3D space. The radiation patterns of the two specific monopole antennas within the 2 × 2 MIMO array are described by the functions ${\vec{R}}_{j}\left(\theta ,\varphi \right)$ and describe the radiation characteristics of the 2 × 2 MIMO dual-band monopole antenna array when excitation is applied to port-1. The ECC results are shown in Figure 17. The simulated and measured *ECC*12 continuously stayed below 0.01 at the 2.4 GHz IEEE 802.11 band (2.04 to 3.25 GHz) and the 5.8 GHz IEEE 802.11 band (5.37 to 7.38 GHz), which indicates good isolation between the antenna elements, according to analysis. The DG, another key parameter of MIMO systems, can be determined by measuring the improvement in single antenna or signal-to-noise ratio (*SNR*) due to the use of multiple antennas.
(2)$$DG=10\sqrt{1-{\left|\rho \right|}^{2}}$$

The simulated and measured *DG*12 values were approximately 10 at the 2.4 GHz band (2.04 to 3.25 GHz) and 5.8 GHz band (5.37 to 7.38 GHz), as depicted in Figure 17. It is noted that ECC and DG computations were obtained from s-parameters, offering important information about the performance features of the 2 × 2 MIMO dual-band monopole antenna system. The outcomes demonstrate the strong functionality of the proposed MIMO dual-band monopole antenna design, exhibiting effective isolation among components and noteworthy diversity gain in the designated frequency bands. While the consistent DG values of approximately 10 demonstrate the system’s capacity to improve signal-to-noise ratio and UAV communication dependability in various operational conditions, the low ECC values confirm little interaction between antenna parts.

This study presented a noteworthy dual-band monopole MIMO antenna that offers numerous advantages compared with previously published work shown Table 2. This work aims to improve the bandwidth, isolation, and radiation coverage using a dual-band monopole MIMO antenna array for UAV communication. At 23 dB and 19 dB, the proposed work achieved excellent isolation at 2.4 GHz and 5.8 GHz compared to previous reports references from [40,41,42,43,44,45,46,47,48,49], respectively. Only reference [50] showed comparable isolation, while other references ranged from 10 dB to 19.3 dB. However, this design has a lower gain compared with the proposed work. The proposed design obtained an impressive bandwidth of 2.04–3.25 GHz and 5.37–7.38 GHz, notably wide compared to reported design references from [40,41,42,43,44,45,46,47,48,49,50,51]. The proposed work presents a well-balanced approach, offering wide bandwidth, high isolation, and moderate gain at 2.4 GHz and 5.8 GHz. The trade-off appears to be in gain, likely balanced against the wide bandwidth and high isolation achieved for the UAV communication system.

Future work will be dedicated to continued development of the dual-band MIMO monopole antenna array. An important improvement strategy is the further optimization of the design to allow for the UAV structural variations. This will be accompanied by scaling the MIMO antenna array to smaller UAVs to keep it optimizable and to optimize the scale for large platforms to allow for integration with other onboard systems. Moreover, future work will consider the antenna’s performance in different testing conditions, including areas with obstacles, such as buildings, dense forests, and other similar situations that may affect signal reflection and diffraction.

## 4. Conclusions

In this manuscript, the proposed dual-band monopole microstrip antennas that operate at both 2.4 GHz band (2.04 to 3.25 GHz) IEEE 802.11b and 5.8 GHz band (5.37 to 7.38 GHz) IEEE 802.11a simultaneously from a single feed point were used in a 2 × 2 MIMO configuration. The proposed MIMO array was designed, fabricated, and tested. Both measured and simulated results demonstrated that the antenna performed effectively in 2.4 GHz and 5.8 GHz, demonstrating good impedance matching, typical monopole radiation patterns, and diversity parameters. Furthermore, an equivalent circuit model for the proposed design was presented, which provided results that closely aligned with the simulations. The MIMO monopole antenna array has a low profile. The MIMO antenna array achieved more than 19 dB isolation at both bands. The total efficiency was observed to be more than 89% at dual bands. This MIMO setup enhanced the UAV’s communication range and versatility with directional coverage across both bands. This work demonstrates a uniquely optimized antenna array well-suited for integration on small UAV platforms to address 5G communication demands.

## Figures and Tables

**Figure 1 sensors-24-05913-f001:**
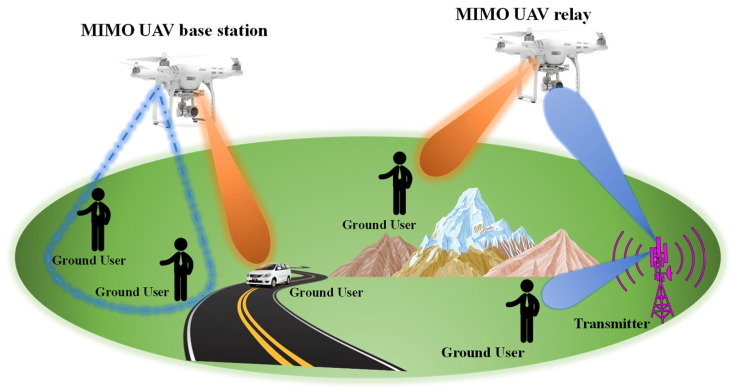
5G services are delivered to ground users by aerial base stations.

**Figure 2 sensors-24-05913-f002:**
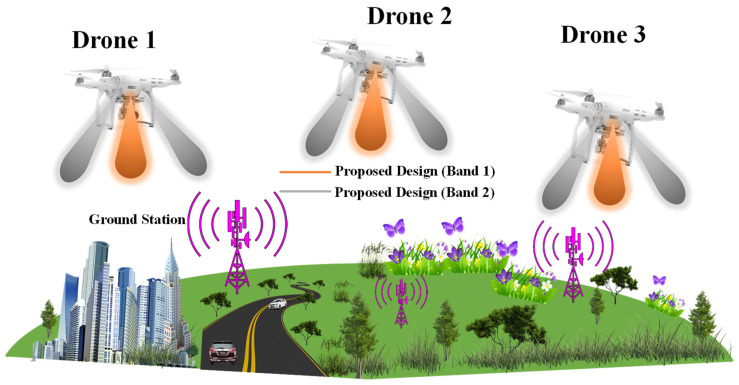
Radiation coverage analysis for a versatile UAV antenna design.

**Figure 3 sensors-24-05913-f003:**
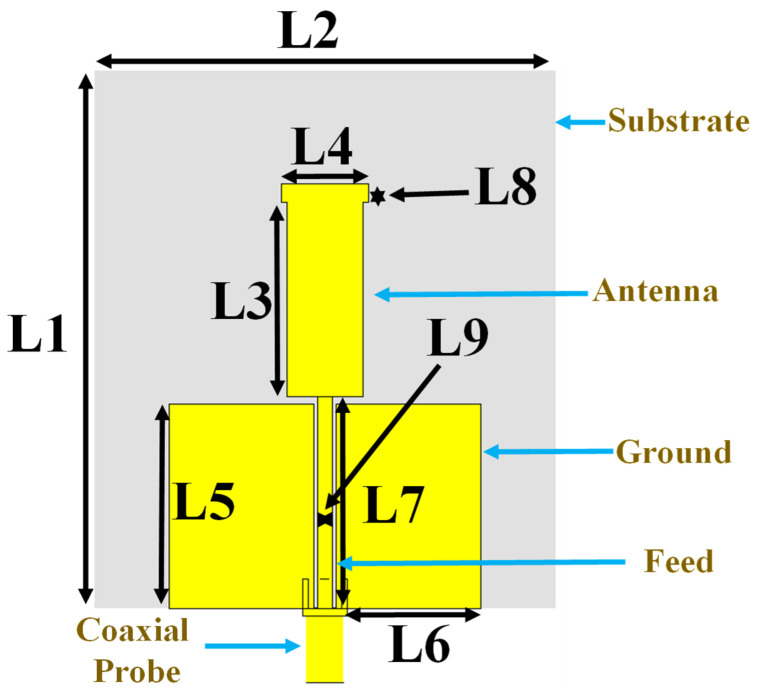
Dual-band monopole antenna.

**Figure 4 sensors-24-05913-f004:**
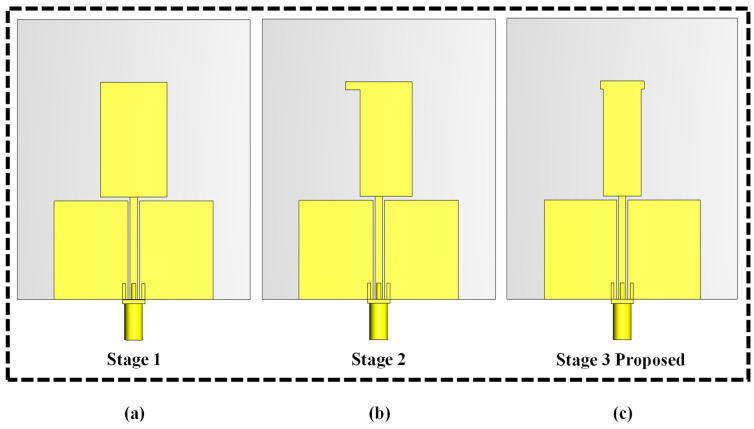
Evaluation of the UAV monopole dual-band antenna structure: (**a**) antenna 1, (**b**) antenna 2, and (**c**) the proposed antenna.

**Figure 5 sensors-24-05913-f005:**
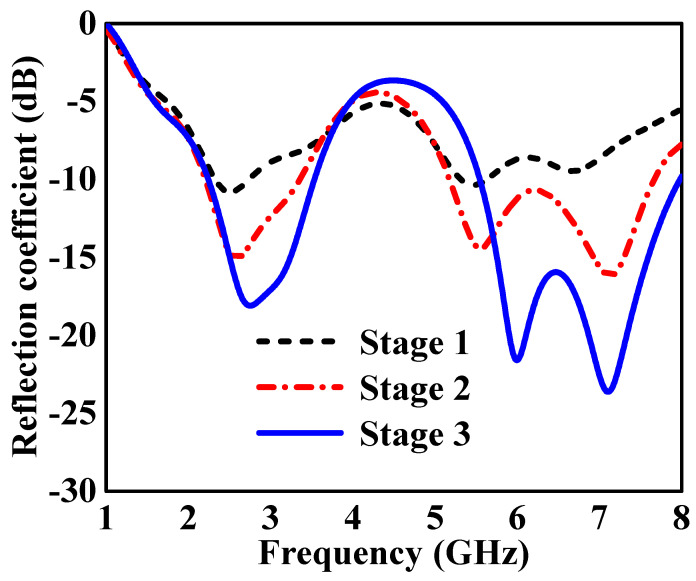
S-parameter S11 of the evaluation structures is shown in Figure 4.

**Figure 6 sensors-24-05913-f006:**
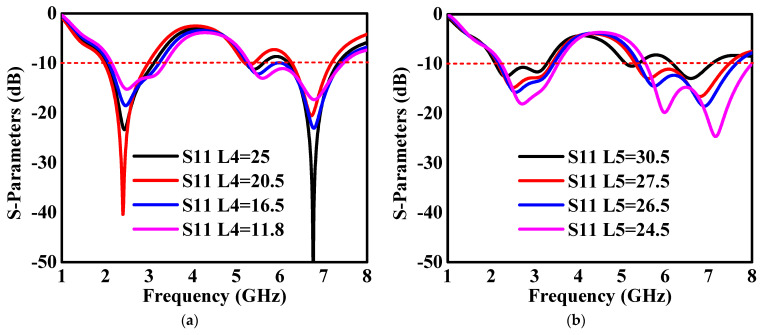
S-parameters of dual-band monopole antennas: (**a**) parametric variation in L4 and (**b**) parametric variations in L5.

**Figure 7 sensors-24-05913-f007:**
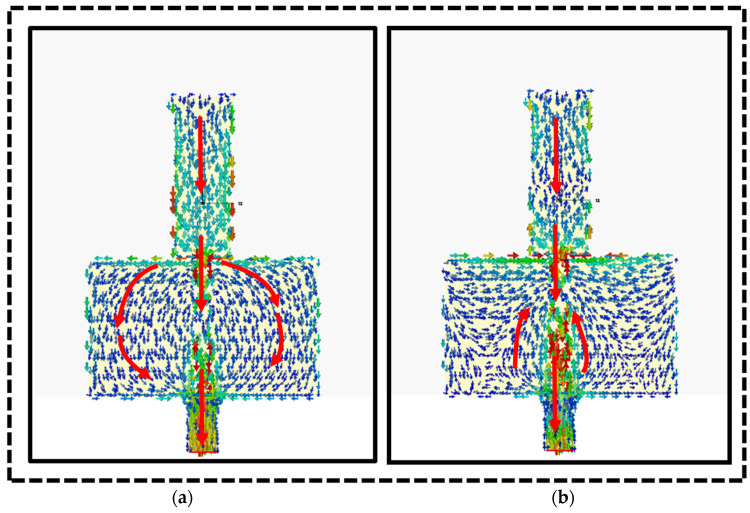
Surface current t distribution of the dual-band monopole antenna at (**a**) 2.4 GHz and (**b**) 5.8 GHz.

**Figure 8 sensors-24-05913-f008:**
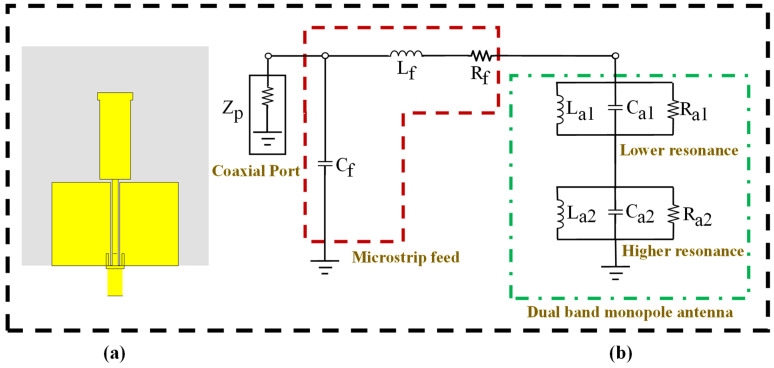
Dual-band monopole antenna design: (**a**) CST design and (**b**) ADS design equivalent circuit model, Zp = 50 Ω, Cf = 1.263 pH, Lf = 1.263 nH, Rf = 5.88 Ω, La1 = 2.12 nH, Ca1 = 2.157 pH, Ra1 = 58.37 Ω, La2 = 0.8168 nH, Ca2 = 0.941 pH, Ra1 = 61.62 Ω.

**Figure 9 sensors-24-05913-f009:**
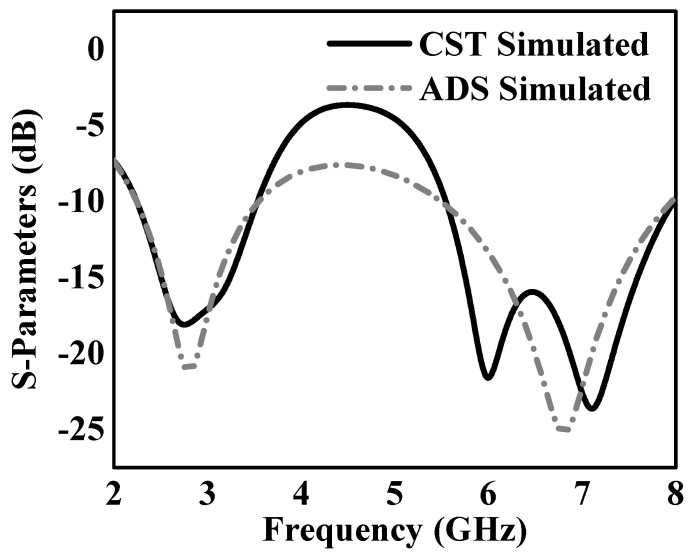
Simulated CST dual-band monopole antenna and ADS equivalent circuit S11 outcomes.

**Figure 10 sensors-24-05913-f010:**
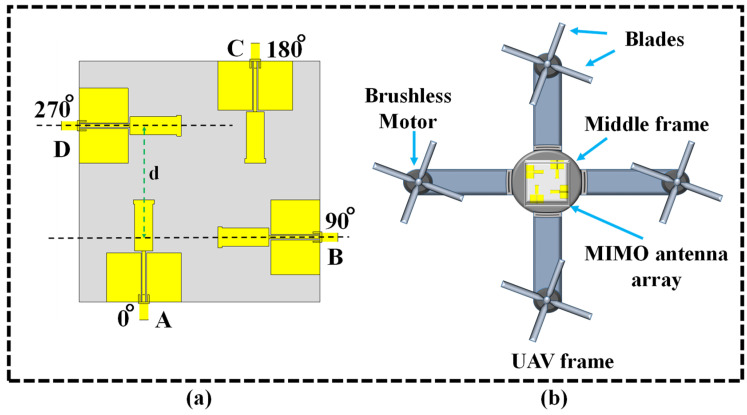
UAV model and MIMO antenna array schematic: (**a**) 2 × 2 dual-band monopole MIMO antenna array and (**b**) UAV model with four monopole MIMO antenna arrays.

**Figure 11 sensors-24-05913-f011:**
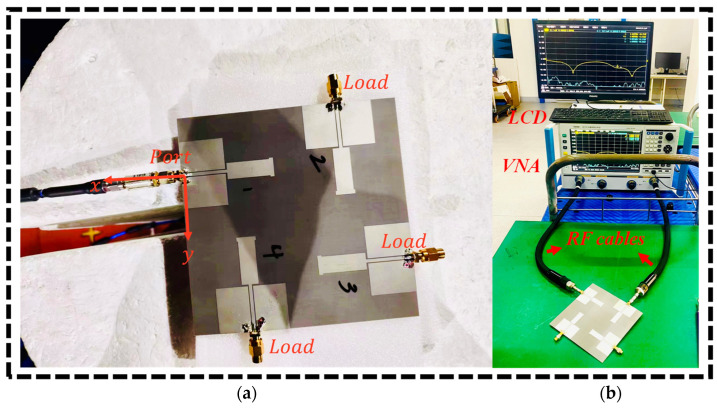
Fabricated prototype of the 2 × 2 MIMO monopole antenna array (**a**): one port is connected with an RF cable and the other ports are connected with (**b**) load measurements of s-parameters using VNA.

**Figure 12 sensors-24-05913-f012:**
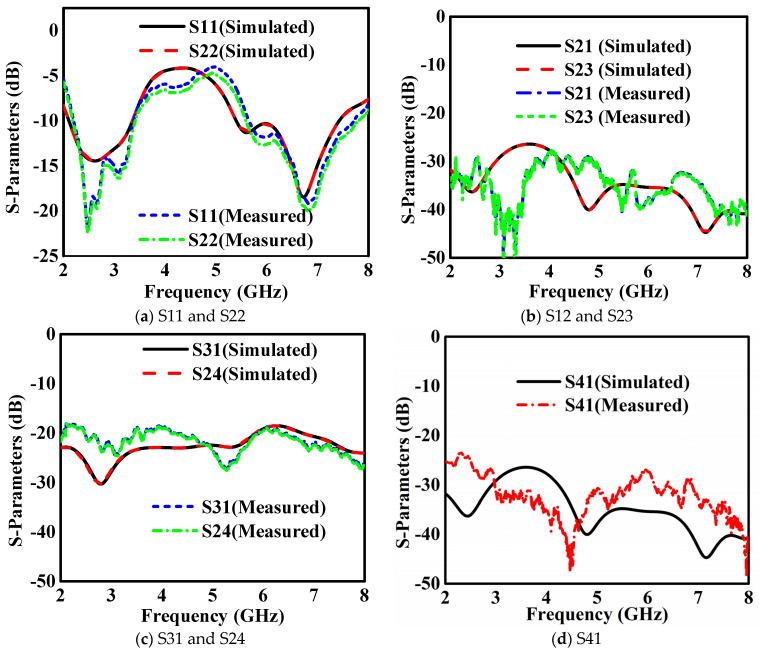
Simulated and measured s-parameters of the 2 × 2 dual-band monopole antennas.

**Figure 13 sensors-24-05913-f013:**
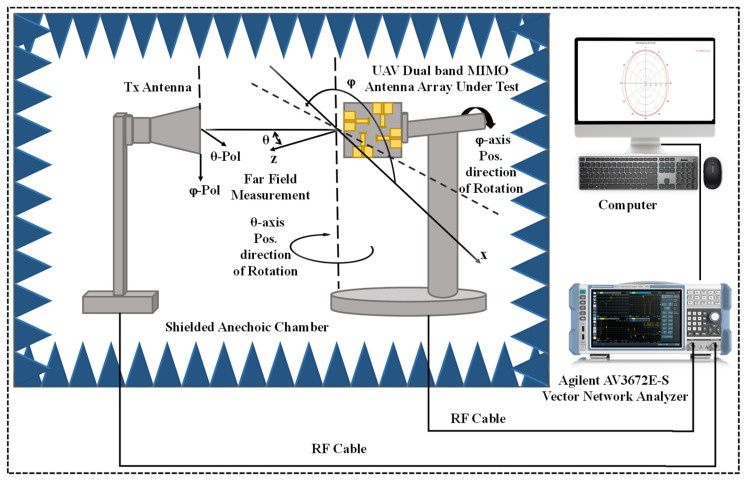
2 × 2 MIMO monopole dual-band array measurement setup.

**Figure 14 sensors-24-05913-f014:**
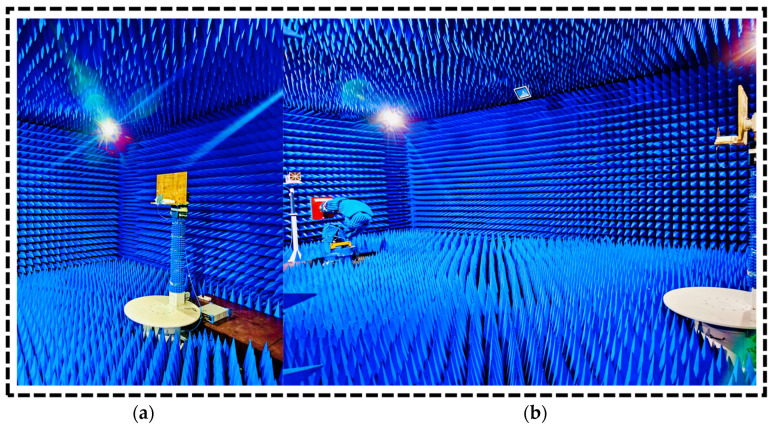
2 × 2 MIMO monopole dual-band array positioned in an anechoic chamber for radiation pattern testing: (**a**) array mounting and (**b**) far-field measurement.

**Figure 15 sensors-24-05913-f015:**
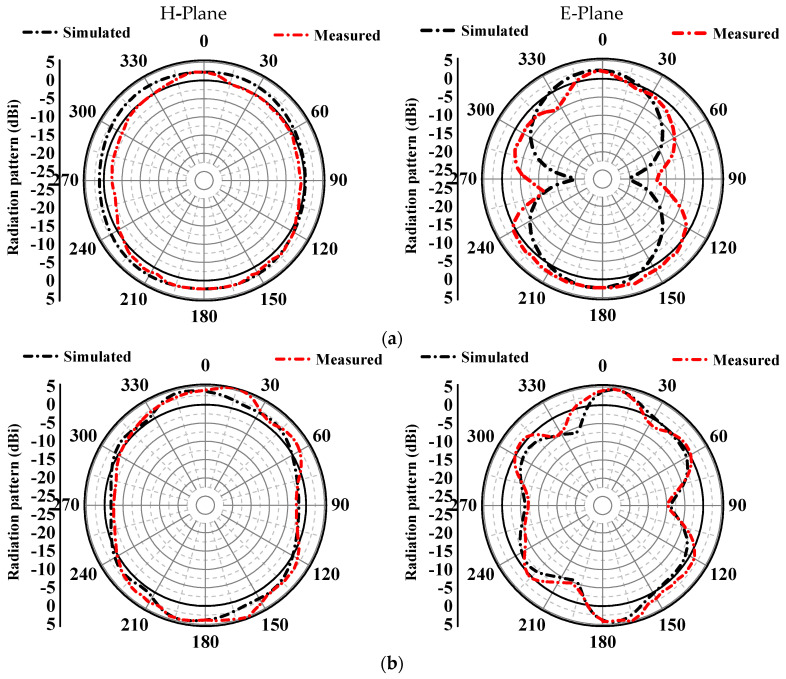
Simulated and measured radiation pattern of the dual-band monopole antenna at the H-plane and E-plane at (**a**) 2.4 GHz and (**b**) 5.8 GHz.

**Figure 16 sensors-24-05913-f016:**
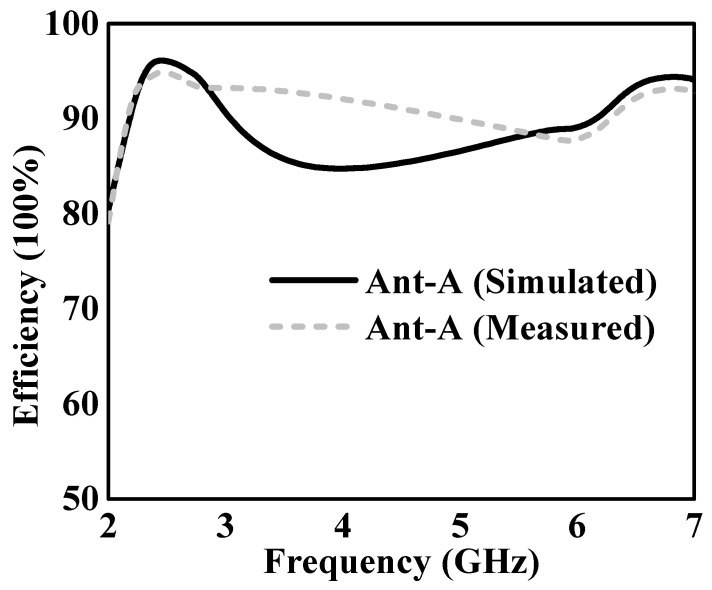
Simulated and measured total efficiency of Ant-A.

**Figure 17 sensors-24-05913-f017:**
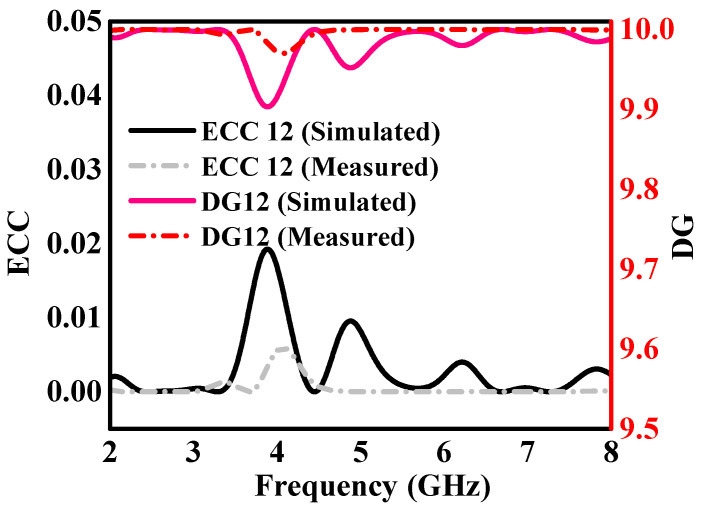
Simulated and measured diversity performance (ECC and DG) of the 2 × 2 MIMO dual-band monopole antenna array.

**Table 1 sensors-24-05913-t001:** Parameters ofthe dual-band monopole antenna.

Parameter	Value (mm)	Parameter	Value (mm)
L1	72.50	L6	19.48
L2	63	L7	28.50
L3	26.20	L8	2.50
L4	11.80	L9	3.00
L5	27.50		

**Table 2 sensors-24-05913-t002:** Performance of proposed MIMO design compared with previously published designs.

Ref.	Central Frequency (GHz)	Substrate Material	No. of Ports	Bandwidth (GHz)	Isolation (dB)	Gain (dBi)
[40]	5.6	FR-4	2	5.5–6.08	14	4.4
[41]	6.44	FR-4	2	5.71–8.2	15	3.8
[42]	5.7	FR-4	3	5.5–6.1	18	2.85
[43]	0.896	Rogers RT5880	1	0.758–1.034	Nil	1.8
[44]	5.7	FR-4	4	5.6–5.8	15.4	1.41
[45]	5.2	FR-4	4	5.05–5.35	15	1.40
[46]	3.7	FR-4	2	3.3–4.2	15	2.5
[47]	28	Ro 4350 B	4	25.5–29.6	10	8.3
[48]	2.4, 5.8	FR-4	2	2.25–2.9, 5.05–6.025	19.3	3.8
[49]	2.45, 5.5	FR4-epoxy	2, 4	2.4–2.5, 4.9–5.725	12, 12	0.5, 2.4
[50]	2.4, 5.1	Nil	2	2.3–2.5, 5–5.2	20, 20	1.28, 2.1
[51]	2.44, 5.5	FR-4	2	2.4–2.48, 5.15–5.83	15	3.1
This work	2.4, 5.8	Rogers RT5880	4	2.04–3.25, 5.37–7.38	23, 19	2.97, 5.49

## Data Availability

The data supporting this research article are available upon request to the corresponding author.

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
