# Peer review of "Dual-Band Monopole MIMO Antenna Array for UAV Communication Systems"

_sensors, 2024, doi:10.3390/s24185913_

Round 1

Reviewer 1 Report

Comments and Suggestions for Authors

 This work presents a dual-band monopole MIMO antenna array designed to address the key requirements for UAV platforms. The proposed dual-band monopole antenna demonstrates one of the widest impedance bandwidths 46.15% at 2.4 GHz IEEE 802.11 and 31.85% at 5.8GHz IEEE 802.11 on a thin 0.0064λo substrate while achieving high transmission efficiency. I have the following comments to further improve the article as follows:

1.      In the abstract, please condense the main elements more concisely. Concentrate on the primary accomplishments and outcomes without going into details that should be included in the main text.

2.      The authors should avoid using an excessive amount of specialized terminology in the introduction and abstract. Provide clear explanations of intricate terminology and ideas to ensure that the document is easily understandable by a wider range of readers.

3.      The paper's beginning lacks a thorough literature review. To emphasize the originality of your work, incorporate more up-to-date research and evaluate current solutions thoroughly. The authors can also consider works such as “A Wideband High Isolated Microstrip MIMO Circularly Polarized Antenna Based on Parasitic Elements,” “Angularly Stable Band Stop FSS Loaded MIMO Antenna with Enhanced Gain and Low Mutual Coupling,” and so on.

4.      The labeling and descriptions in the captions of Figures 1, 2, and 10 should be improved for better clarity. The text should sufficiently elucidate all components in the figures, and the quality of the photos should be enhanced to enhance legibility.

5.      The methodology is deficient in providing adequate specificity regarding the simulation setup and the parameters employed. Provide a more detailed and thorough explanation of the experimental methods and the specific apparatus utilized.

6.      The comparative analysis is lacking in strength. There is a deficiency in conducting comparison analysis with the current antenna designs. Include a part that conducts a comparative analysis of the performance of your suggested antenna with other cutting-edge designs to showcase its superiority.

7.      A detailed comparison with other state-of-the-art approaches is completely missing. The authors should add a proper table for comparison. Without a proper comparison, how can the novel researcher differentiate between the existing work and your work? Please look at Table 2 in the paper “Designed Circularly Polarized Two-Port Microstrip MIMO Antenna for
WLAN Applications” and then add a comparison table in your paper accordingly.

8.      The results section should offer a more comprehensive examination of the data presented.

9.      The conclusion is excessively concise and fails to adequately encapsulate the primary discoveries or their wider significance.

10.   The authors should verify that all citations are current and pertinent. Several citations seem to be obsolete, and you must provide additional up-to-date sources to substantiate your assertions.

Comments on the Quality of English Language

 Extensive editing of English language required.

Reviewer 2 Report

Comments and Suggestions for Authors

There are some concerns as follows:

1. what is the significance of Fig. 2? Did you analyzed the radiation coverage placing your designed antenna on three drones?

2. The optimization process of the antenna design is not clearly explained in the paper, leaving readers uncertain about the methods or mathematical models used. For a robust understanding, the authors should clarify whether they employed techniques such as genetic algorithms, particle swarm optimization, or another method to refine the antenna's physical parameters. Additionally, the inclusion of any mathematical formulations or simulations that guided the optimization process would greatly enhance the technical clarity and depth of the work. This would also help in understanding the rationale behind the design choices and how they were optimized to achieve the desired performance.

3. The novelty of this work is not explicitly stated, which may lead to confusion regarding its unique contributions. To strengthen the paper, the authors should clearly articulate the innovative aspects of their design, such as any advancements in performance, materials, or methodologies compared to existing solutions. Additionally, discussing the limitations of the design, such as potential challenges in fabrication, scalability, or performance in specific environments, would provide a more balanced and comprehensive view of the work's significance and areas for future improvement.

4. incorporate Channel Capacity Loss (CCL): Measures the loss in capacity due to correlation between antennas. and Total Active Reflection Coefficient (TARC): Represents the reflection of the active signals when all antennas are in use, and Diversity Gain: Refers to the improvement in signal reliability by using multiple antennas. 

5. In the introduction, the authors discuss numerous papers but fail to include any articles from 2024. It is recommended that the authors review recent publications, including the following: https://ieeexplore.ieee.org/abstract/document/10561872,   and https://ieeexplore.ieee.org/abstract/document/10088497 and  discuss them in the introduction section. It is need to include the strengths and limitation of the reviewed paper in the literature review part.  

6.  revise the explanation of parametric study and equivalent circuit of the proposed antenna to make it more understandable and technically sound.

7. How the antenna is suitable for UAV applications? since the antenna is placed at the drone so you may introduce some words about wireless power transfer concept in your discussion. read carefully. https://onlinelibrary.wiley.com/doi/full/10.1002/eng2.12951 to get the concept and use it.

 8, Figure 15 needs to be replaced with a higher-quality version to enhance its clarity and visibility. A clearer image will help in better illustrating the key points and findings presented in the figure, ensuring that readers can fully understand the data and its implications. Improving the resolution or quality of Figure 15 is crucial for conveying the information accurately and effectively.

9. provide a comparison table with some relevant recent works

10.  Improve flow of information through out the paper and check grammatical error and typos

Comments on the Quality of English Language

Moderate

Round 2

Reviewer 1 Report

Comments and Suggestions for Authors

I have no more comments. 

Reviewer 2 Report

Comments and Suggestions for Authors

Authors have addressed all the comments properly. Now it can be published.

Comments on the Quality of English Language

Okay